# The SR Splicing Factors: Providing Perspectives on Their Evolution, Expression, Alternative Splicing, and Function in *Populus trichocarpa*

**DOI:** 10.3390/ijms222111369

**Published:** 2021-10-21

**Authors:** Xijuan Zhao, Lingling Tan, Shuo Wang, Yirong Shen, Liangyu Guo, Xiaoxue Ye, Shenkui Liu, Ying Feng, Wenwu Wu

**Affiliations:** 1State Key Laboratory of Subtropical Silviculture, School of Forestry and Biotechnology, Zhejiang Agriculture and Forestry University, Hangzhou 311300, China; zhao_xijuan@163.com (X.Z.); linglingtan_0108@163.com (L.T.); wangshuo@zafu.edu.cn (S.W.); Yirong_Shen@outlook.com (Y.S.); guo_liangyu2021@163.com (L.G.); shenkuiliu@nefu.edu.cn (S.L.); 2Institute of Tropical Biosciences and Biotechnology, Chinese Academy of Tropical Agricultural Sciences, Haikou 571101, China; xiaoxueyexx@gmail.com; 3Key Laboratory of Nutrition, Metabolism and Food Safety, Shanghai Institute of Nutrition and Health (SINH), Chinese Academy of Sciences (CAS), Shanghai 200032, China

**Keywords:** *Populus trichocarpa*, serine/arginine-rich (SR) protein, alternative splicing, abiotic stress, *PtSCL30*

## Abstract

Serine/arginine-rich (SR) proteins are important splicing factors in plant development and abiotic/hormone-related stresses. However, evidence that SR proteins contribute to the process in woody plants has been lacking. Using phylogenetics, gene synteny, transgenic experiments, and RNA-seq analysis, we identified 24 *PtSR* genes and explored their evolution, expression, and function in *Popolus trichocarpa*. The *PtSR* genes were divided into six subfamilies, generated by at least two events of genome triplication and duplication. Notably, they were constitutively expressed in roots, stems, and leaves, demonstrating their fundamental role in *P. trichocarpa*. Additionally, most *PtSR* genes (~83%) responded to at least one stress (cold, drought, salt, SA, MeJA, or ABA), and, especially, cold stress induced a dramatic perturbation in the expression and/or alternative splicing (AS) of 18 *PtSR* genes (~75%). Evidentially, the overexpression of *PtSCL30* in *Arabidopsis* decreased freezing tolerance, which probably resulted from AS changes of the genes (e.g., *ICE2* and *COR15A*) critical for cold tolerance. Moreover, the transgenic plants were salt-hypersensitive at the germination stage. These indicate that *PtSCL30* may act as a negative regulator under cold and salt stress. Altogether, this study sheds light on the evolution, expression, and AS of *PtSR* genes, and the functional mechanisms of *PtSCL30* in woody plants.

## 1. Introduction

Alternative splicing (AS) is an important mechanism in the regulation of gene expression in eukaryotes, which enhances transcriptome and proteome diversity [1,2]. Over 95% of human protein-coding genes can be alternatively spliced to produce multiple transcripts, such as *KCNMA1* can produce more than 500 mRNA isoforms [3,4]. In plants, about 83% and 73% of intron-containing genes undergo AS in *Arabidopsis thaliana* and *Oryza sativa*, respectively [5,6]. There are mainly five different types of AS events, including exon-skipping (ES), intron retention (IR), mutually exclusive exons (MXE), alternative 5′ splice site (A5SS) and 3′ splice site selection (A3SS) [7]. IR is a major mode of AS in plants, whereas ES is a predominant mode in animals [8,9,10]. The importance of AS has been clearly manifested by the genetic hereditary diseases caused by splicing defects [11,12].

Due to a sessile life form, plants need unique adaptive developmental and physiological strategies to cope with environmental perturbations. AS is emerging as an important process affecting plant development and tolerance to biotic and abiotic stresses. AS can regulate transcriptome and proteome plasticity to respond rapidly to environmental stresses by adjusting the abundance of the functional transcripts of the stress-related genes, such as protein kinases, transcription factors, splicing regulators, and pathogen-resistance genes [13]. For example, hundreds of genes, such as novel cold-responsive transcription factors and splicing factor/RNA-binding proteins, showed rapid AS changes in response to cold (called ‘early AS’ genes) [14]. More than 6,000 genes were reported to undergo changes of AS patterns under salt stress [15,16]. In addition, AS is also involved in a range of other functions, such as photosynthesis, circadian clock, flowering time, and metabolism [17,18,19,20].

Pre-mRNA splicing processing is catalyzed by a spliceosome, a large flexible RNA-protein complex consisting of five small nuclear ribonucleoprotein particles (snRNPs) and numerous types of non-snRNP proteins [21,22]. Serine/arginine-rich (SR) proteins, the major regulators in the splicing of pre-mRNAs, are evolutionarily conserved splicing factors [23,24]. In plants, SR proteins were defined as one or two N-terminal RNA recognition motifs (RRMs) followed by a downstream RS domain of at least 50 amino acids with over 20% SR or RS dipeptide [25]. SR family proteins have been identified in many plant species, such as green algae, moss, and various flowering plants. The number of *SR* genes varies among different species; for example, there are 18 members in *Arabidopsis*, 22 in *O. sativa*, 21 in *Dimocarpus longan Lour*, 40 in *Triticum aestivum*, and 18 in *Brachypodium distachyon* [26,27,28,29]. Plant SR proteins can be classified into six subfamilies, including SR, SC, RSZ, RS, SCL, and RS2Z. The SR, SC, and RSZ subfamilies have orthologs in mammals, while the RS, SCL, and RS2Z subfamilies are unique to plants with novel structural features [25]. RS subgroup members have two RRM domains, and the second RRM domain lacks the SWQDLKD signature, which is a characteristic of SR-subfamily proteins. RS2Z subfamily members have two Zn-knuckles and one RS domain, followed by an SP-rich region. SCL-subfamily members have a single RRM domain followed by an RS domain, and possess a short N-terminal extension that contains multiple RS and SP dipeptides [30,31].

Plant *SR* genes are involved in various plant growth and development processes. The overexpression of *AtSRp30* resulted in a delayed transition from the nutrition to reproductive periods, prolonged life cycle, and increased individual size [32]. The loss of *Arabidopsis* SC35/SCL proteins led to multiple effects on plant morphology and development, such as serrated leaves and later flowering [33]. Additionally, plant *SR* genes can be alternatively spliced, and their splicing patterns are affected by various developmental and environmental signals. For example, the overexpression of *RSZ36* and *SRp33b* can change the splicing patterns of *RSZ36* and *SRp32* in rice, respectively [34]. High temperatures may increase the expression of the active isoforms of *SR30* but reduce the active isoform of *SR34* in *Arabidopsis* [35]. Moreover, *SR* genes may allow functional redundancy in the processes of plant growth and development. In *Arabidopsis*, the *sc35-scl* quintuple mutant (*scl28 scl30 scl30a scl33 sc35*) exhibited the obvious phenotypes of serrated rosette leaves and late-flowering, while no obvious morphological alterations were observed in the double or triple mutants [33].

*Populus trichocarpa* is a model species of woody plants in which to study the repertoire of biological processes in trees [36]. However, no systematic analysis and functional characterization of the *SR* gene family has been reported in this woody plant. In this study, we identified the *PtSR*-family genes of *P. trichocarpa* and performed comprehensive analyses of the identified *PtSR* genes. In addition to their evolution, we also investigated the expression profiles and AS events of *PtSR* genes in different tissues and under various stresses. Moreover, we demonstrated the function of a plant-specific *SR* gene *PtSCL30* through the overexpression in *Arabidopsis* and RNA-seq analyses. Our results provide a resource for *SR* genes with respect to their evolution, expression, and alternative splicing in *P. trichocarpa*, contributing to the knowledge of RNA-splicing in trees’ development and responses to environmental stresses.

## 2. Results

### 2.1. Identification of PtSR Family Genes and Their Characteristics

Firstly, we searched for the homologs of *Arabidopsis* SR proteins in *P. trichocarpa* genome by the BLASTP program [37]. Taking account of the definition of the RS domain, which has at least 50 amino acids sharing over 20% RS content by consecutive RS or SR dipeptides in plants [25], we screened the homologs to obtain a total of 24 PtSR proteins. We then assessed the basic characteristics of the 24 PtSR proteins; for example, their molecular weights (*M*_W_s) ranged from 20.46 to 34.56 kDa, with an average value of 29.3 kDa (Appendix A). Of note, all the proteins had an extremely high isoelectric point (pI) between 9.9 and 11.6, and, consequently, were highly cationic at neutral or acid pH. This is supported by the fact that SR proteins can bind the negatively charged RNA in nuclei [38]. Additionally, based on the grand average of hydropathy (GRAVY) values, all the proteins were predicted to be hydrophilic between −1.772 and −0.881, supporting the soluble nature of PtSR proteins. Detailed characteristics of the assessed *PtSR*-family genes are presented in Appendix A.

### 2.2. Phylogenetic and Architectural Analysis of PtSR Family Genes

Since *SR* genes have been widely studied in *Arabidopsis*, we selected *SR* genes from *Arabidopsis* as a reference and constructed a phylogenetic tree based on the full-length alignment of the SR proteins in the species *Arabidopsis* and *P. trichocarpa* (Figure 1A). PtSR proteins were classified into the six known subfamilies, SR, SC, RSZ, RS, SCL, and RS2Z. This result agreed well with those from *Arabidopsis* [39], indicating that the *SR* gene family was highly conserved, at least in the dicots. As compared to animals, ~50% of *PtSR* genes were plant-specifically evolved *SR* genes, including the previously reported RS, SCL, and RS2Z subfamilies [25].

Gene exon/intron structure diversity is one of the possible mechanisms for explaining the evolution of multiple gene families, to which end, we further analyzed the structures of the *PtSR* genes (Figure 1B). Observably, the *PtSR* genes were interrupted by multiple introns, ranging between 4 and 13, and, expectedly, the clustered *PtSR* genes showed similar exon–intron structures and shared a recent common ancestor. In detail, the same subfamily had a very similar number of introns (Figure 1B). For example, the *SR*-subfamily genes had the most introns, ranging between 12 and 13, while the *RS2Z*-subfamily genes had the same number (six) of introns. This showed that the subfamilies of the *SR*-family genes were highly conserved after their divergence from their nearby subfamilies.

In the case of PtSR-protein domains, we retrieved the conserved protein domains based on the annotated domains from the Pfam database [40]. Two types of homolog-based domains were finally identified, including the RRM and zf-CCHC domains (Figure 1C). Expectedly, all the PtSR proteins had at least one RRM and RS domain. Meanwhile, some differences were also found between the subfamilies, such as the one and two zf-CCHC domains, respectively, in the RSZ and RS2Z subfamilies. Finally, and noteworthy, among the six subfamilies in *P. trichocarpa*, SCL was the largest, followed by SR; whereas, in *Arabidopsis*, three subfamilies, SCL, SR, and RS, were very close in number (Figure 1D). Next, we mapped the detailed expansion of these subfamilies.

### 2.3. The Expansion History of the PtSR Gene Family in P. trichocarpa

To investigate the evolution of *PtSR* gene family, we determined their chromosomal distributions and gene-duplication types. The *PtSR* genes were distributed unequally to *P. trichocarpa* chromosomes (the outer circle in Figure 2A). Three *PtSR* genes (*PtRSZ22*, *PtSCL25* and *PtRS2Z32*) were located on chromosome (Chr) 6, followed by two *PtSR* genes on Chrs 2, 5, 8, 10, 14 and 16, respectively. Of note, except for the SCL-subfamily genes (e.g., *PtSCL28* and *PtSCL30*) being located in the same chromosomes, genes from the same *PtSR* subfamily were mainly distributed to different chromosomes, declining the possibility of generating *PtSR* genes by tandem or proximal duplications.

To determinate molecular mechanisms generating the *PtSR*-family genes, we traced their expansion history and found a total of 16 collinear gene blocks, including 21 *PtSR* genes (the inner color lines in Figure 2A). This finding showed that the whole- and/or segmental-genome duplication pattern was the dominant molecular mechanism generating the *PtSR* genes. To date the events of the 16 collinearity gene blocks, we calculated the synonymous substitution rate (Ks) of the duplicated gene pairs (Appendix A) and found the blocks could be mainly classified into two categories (Figure 2B). The first category included six gene pairs (shown in blue lines and dots in Figure 2A,B), and their Ks values varied between 1.5649 and 2.1702 (Figure 2B, Appendix A), which were around the whole-genome triplication event (i.e., γ) in the recent common ancestor of *P. trichocarpa* and *Arabidopsis* [41]. The other category included ten gene pairs (shown in red lines and dots in Figure 2A,B), and their Ks values varied between 0.2164 and 0.6339 (Figure 2B, Appendix A), and were concentrated around the most recent whole-genome duplication event of *P. trichocarpa* [42]. This recent duplication event was successful in replicating the genes of the SCL and SR subfamilies (Figure 2), explaining well the existing larger number of the two subfamilies in *P. trichocarpa* than in *Arabidopsis* (Figure 1D). Together, the two categories demonstrated at least two expansion stages of the *SR* gene family through genome polyploidization, which provided the dominant molecular mechanism producing the existing *PtSR* genes in *P. trichocarpa*.

Accompanied by the duplications of *SR* genes, their flanking promoters were also copied. Next, we investigated the enriched functions and the promoter *cis*-acting elements of the *PtSR* genes.

### 2.4. GO Term Enrichment and Promoter Cis-element Analysis of PtSR Genes

We performed GO term-enrichment analysis of *PtSR* genes to investigate the molecular functions and biological processes that *PtSR* genes might participate in. The result showed that *PtSR* genes could participate in various biological processes, such as spliceosome assembly, RNA splicing, mRNA export, the regulation of metabolic processes, the response to stress, and the regulation of gene expression (Figure 3A). This indicated that *PtSR* genes not only act as splicing factors for RNA-splicing and metabolism processes at the post-transcriptional level, but also might be involved in the diverse and complicated regulation of gene expression at the transcriptional level.

To identify the *cis*-acting elements in the promoters of *PtSR* genes, we analyzed the 2-kb sequences upstream of the translation-start sites of *PtSR* genes in PlantCARE [43] (Figure 3B). Firstly, there were well-known housekeeping *cis*-acting elements in the promoters of all the *PtSR* genes, such as TATA-box and CAAT-box. Also, some *cis*-acting elements were enriched in response to phytohormones, such as abscisic acid (ABA), methyl jasmonate (MeJA), salicylic acid (SA), and gibberellin (GA). Furthermore, the *cis*-acting elements were also enriched in response to abiotic stresses, such as low-temperature, drought, defense and stress, and anaerobic induction (Figure 3B, Appendix A). The housekeeping and hormone/abiotic-responded *cis*-elements, together, demonstrated that *PtSR* genes are probably expressed in constitutive regulation, but also in response to hormone/abiotic stresses, and we next studied their expressions in *P. trichocarpa* tissues and under hormone/abiotic stresses.

### 2.5. Constitutive and Abundant Expression Patterns of PtSR Genes in P. trichocarpa

To investigate the expression profiles of *PtSR* genes, we analyzed the expression levels of *PtSR* genes by RNA-seq data in *P. trichocarpa* tissues including roots, stems, and leaves. According to RNA-seq data, we found that *PtSR* genes mainly exhibited constitutive expression profiles in all of the three tissues (Figure 4A), and the relative expression levels of *PtSR* genes were significantly higher than the background-expressed genes (Figure 4B). Of note, many *SR* genes showed an observably higher expression than the well-known housekeeping gene (Figure 4A). The results together suggested that *PtSR**s* were constitutively and abundantly expressed in different tissues of *P. trichocarpa*.

### 2.6. Perturbation in PtSR Gene Expressions by Hormones and Abiotic Stresses

As described above, the *cis*-element analysis indicated that *PtSR* genes might also be involved in abiotic and hormone stresses. In this section, we analyzed the expression profiles of *PtSR* genes under hormone treatments (SA, MeJA, and ABA) and abiotic stresses (cold, drought, and salt). Of the 24 *PtSR* genes, 17 (~70%) were differentially expressed under at least one of the six stresses. In the hormone treatments, *PtRSZ22a* was significantly induced by ABA, *PtSR30* and *PtSR30a* were respondent to SA, and *PtSR30a* and *PtRS2Z32* responsed to MeJA (Figure 5A). Among them, *PtSR30a* was induced by both hormones. In contrast, 12 (50%) of the *PtSR* genes were differentially regulated under cold treatment, including one down-regulated versus eleven up-regulated ones (Figure 5). We found that *PtSR* genes showed different responses at different time points of cold stress. Some *PtSR* genes were significantly up-regulated at 7 d of cold stress, such as *PtSR34*, *PtSR35a*, *PtSC35*, and *PtRS40*. Moreover, *PtRSZ22a*, *PtSCL28a*, *PtSCL30*, and *PtRS2Z32* were significantly up-regulated at both time points. Under drought stress, we found that only *PtSCL25* was differentially expressed. Under salt treatment, eight *PtSR* genes were differentially expressed (Figure 5A). Except that *PtRSZ22a* was down-regulated, seven *PtSR* genes including *PtSR30*, *PtSR30a*, *PtRSZ21*, *PtSCL25*, *PtSCL26*, *PtSCL28a*, and *PtRSZ32* were up-regulated uniquely at 7 d salt treatment. Considering that *PtSR* genes were generally upregulated by abiotic/hormone stresses, we suggested an enhancing splicing role of *PtSR* genes under these stresses, especially cold conditions.

### 2.7. Conserved Alternative Splicing of PtSR Genes in P. trichocarpa Tissues and Their Perturbance by Hormones and Abiotic Stresses

The SR pre-mRNAs themselves can also be alternatively spliced in different plant tissues and in response to diverse stress treatments [44]. In order to determine either tissue- and/or stress-specific AS of *PtSR* genes, we performed a detailed analysis of splicing variants in *P. trichocarpa* tissues (e.g., roots, stems, and leaves) under various stresses (e.g., cold, drought, salt, SA, MeJA, and ABA) by RT-PCR (Figure 6). By AS, over 45 transcripts were produced from the 24 *PtSR* genes, resulting in nearly two-fold increased complexity of the *SR* family genes (Appendix A). In detail, some *PtSR* genes showed only one transcript, while the others underwent AS events to generate multiple transcripts and there were stress-specific, e.g., cold-induced AS of *PtSCL33* (Figure 6). Excepting that some transcripts were produced by excluding parts of exon regions, most of the alternative transcripts were produced by IR events. Moreover, the *PtSR* genes with conserved exon/intron structures generally showed conserved AS events, such as in*PtSCL25* and *PtSCL26* (Figure 6). Among the roots, stems, and leaves, the AS genes showed a conserved AS regulation (Figure 6) as well as the conserved expression patterns described above (Figure 4).

Compared to normal condition, abiotic stresses could alter their AS variants (Figure 6). Of the abiotic stresses, cold significantly altered the AS outcomes of seven *PtSR* genes (i.e., *PtSCL33*, *PtSCL26*, *PtSCL25*, *PtSR30*, *PtSR30a*, *PtSR34a*, and *PtRS41*), together likely contributing to an increase of the long transcripts (Figure 6). For example, the abundance of THE basic transcript of *PtSR34a* (the short transcript) decreased gradually with the prolongation of treatment time, and the long transcript became more abundant after 7 d of cold treatment. Moreover, a novel long transcript of *PtSCL33* appeared after 24 h cold treatment, and this transcript became more abundant than the basic one after 7 d of cold treatment (Figure 6). In contrast, drought and salt stresses, together, altered only five *PtSRs*’ AS patterns (i.e., *PtSCL26*, *PtSCL25*, *PtSR30*, *PtSR30a*, and *PtRS41*), likely contributing to an increase in the short transcripts. For example, the short transcript abundance of *PtSCL26* was low, while its abundance increased under drought and salt stresses, especially after 7 d of salt treatment. Another example is that the long transcript of *PtSCL25* disappeared under 7 d of drought and 7 d of salt stress, whereas the short transcript continually accumulated. Under hormone treatments, some *PtSR* genes (e.g., *PtSR30a*, *PtSR34a*, *PtSR35*, and *PtRS41*) altered their AS patterns, and the alterations were mainly reflected in the changes in abundances and the disappearance of some transcripts (Figure 6). For example, the long transcript of *PtSR30a* disappeared after the hormone treatments. Moreover, and of importance, we noticed that different plant development stages also affected the AS patterns of *PtSR* genes, such as *PtSR34* and *PtSR35* in the CK controls of different sampling times (Figure 6).

Together, the findings showed a conserved AS regulation as well as their conserved expression patterns among the three tissues, implying the conserved housekeeping roles of *PtSR* genes in the normal growth of *P. trichocarpa* tissues. In contrast, the abiotic/hormone stresses induced a perturbation of AS regulation as well as its expression, and especially, cold resulted in the most changes of AS patterns by IR events, increasing the long transcripts of *PtSR* genes, which might play crucial roles in the cold response. We next exemplified a cold-affected *PtSR* gene to explore the molecular mechanism and functions of *PtSR* genes.

### 2.8. Overexpression of PtSCL30 Decreased the Freezing Tolerance of Arabidopsis

To further explore the function of *PtSCL30*, we transformed *PtSCL30* into *Arabidopsis* and obtained fourteen 35S::*PtSCL30* overexpression (OE) lines, which were confirmed by RT-PCR and qRT-PCR (Appendix A). Finally, three independent stable homozygous 35S::*PtSCL30* overexpression lines (OE2, OE10, and OE16) were selected for further analysis. As described above, *PtSCL30* was significantly upregulated by cold stress, we further investigated the freezing tolerance of 35S::*PtSCL30* OE lines. Under normal conditions (22 °C), no significant difference was observed between the wild-type plants Col-0 and 35S::*PtSCL30* OE lines. However, after freezing treatment (−7 °C), most of the 35S::*PtSCL30* OE lines exhibited freezing-sensitive phenotypes (Figure 7A), and meanwhile, the survival rates of 35S::*PtSCL30* OE lines were significantly lower than Col-0 (Figure 7B). Our results suggested that the overexpression of *PtSCL30* decreased the freezing tolerance of *Arabidopsis* and the underlying molecular mechanism is examined in the next section.

### 2.9. PtSCL30 Overexpression Affected the Alternative Splicing of Hundreds of Genes, including Cold-Responsive Genes

We applied high-throughput sequencing to analyze the alteration of transcriptomes between three-week old 35S::*PtSCL30* OE line (OE10) and wild-type plant (Col-0) under cold stress. Compared to Col-0, we identified 57 DEGs and 206 DASGs in OE10 (Figure 8A). Observably, the overlap of DEGs and DASGs was very limited, showing an independent regulation of gene expression and alternative splicing induced by *PtSCL30* overexpression. However, the number of DASGs was much higher than that of DEGs, indicating that *PtSCL30* might mainly act as a splicing factor to regulate gene splicing under cold stress. We further assessed the AS types of these DASGs occurring in OE10. Of the types, IR was the largest one, followed by ES (Figure 8B). Furthermore, we performed GO enrichment analysis for the *PtSCL30*-affected DASGs (Figure 8C). The results showed that DASGs could participate in regulating of the timing of the meristematic phase transition, response to sucrose, sterol biosynthetic process, response to desiccation, cellular response to oxidative stress, regulation of cellular catabolic process, cold acclimation, and so on (Figure 8C); the strikingly *PtSCL30*-affected network involved in abiotic stress is indicated in red shadow, also in Figure 8C. Of the DASGs, we presented four examples that had been reported to play critical roles in the process of the cold response [45], including *CP29* (*AT3G53460*), *ICE2* (*AT1G12860*), *LHY* (*AT1G01060*), and *COR15A* (*AT2G42540*) (Figure 8D). These cold-responsive genes were differentially regulated in AS under cold stress in *PtSCL30* overexpression as compared with Col-0. In addition to cold response, we noticed that *PtSCL30* can regulate the alternative splicing of genes in response to oxidative and desiccation stresses (Figure 8C). To further explore its role, we examined the phenotypes of *35S::PtSCL30* OE lines under salt and drought stresses.

### 2.10. PtSCL30 Overexpression Were Hypersensitive to Salt Stress

At the germination stage, the 35S::*PtSCL30* OE lines were more sensitive to salt stress than Col-0 (Figure 9A) and the germination rate and cotyledon-greening rate of the 35S::*PtSCL30* OE lines were dramatically lower than for Col-0 (Figure 9B–D), although the difference of root length was not significant (Figure 9E,F). For drought stress, we analyzed the fresh weight and root length of plants after 7 d of vertical growth in 1/2 MS-mannitol medium with mannitol (0 mM, 100 mM, and 200 mM), and found only a subtle difference between Col-0 and 35S::*PtSCL30* OE lines (Figure 9G–I). These results suggested that 35S::*PtSCL30* OE lines were hypersensitive to salt stress at germination stage, whereas the function of *PtSCL30* might be limited in the drought stress.

## 3. Discussion

SR proteins are well-known splicing factors that play important roles in both the assembly of spliceosomes and the regulation of alternative splicing. In plants, *SR* family genes have been widely studied in the model plant *Arabidopsis*. *SR34* (previously named *SR1*) was the first *SR* gene identified in *Arabidopsis* [46]. Then, the *SR* gene family of *Arabidopsis* has been extensively studied, including gene family analysis, variable splicing under stress and hormone treatments, functions of individual *SR* genes, and interaction between *SR* genes and other splicing factors [47,48,49]. In addition, the *SR*-related genes such as *SR45* were also found deepening [50,51].

In this study, we totally identified 24 *PtSR* genes in *P. trichocarpa* and performed comprehensive analyses on their evolution, expression, and functions. In the same fashion as the model plant *Arabidopsis*, *PtSR* genes can also be divided into six subfamilies, demonstrating an ancient origin and the probable conserved functions of the *SR* genes in plants, as members within the same subfamily had similar gene structures and shared conserved motifs, implying that *PtSR* genes in the same subfamily may be functionally redundant. Plants generally have more SR proteins than do animals, which may be due to multiple paralogous gene pairs produced by the expansion of the plant SR family within several rounds of genome replication. *PtSR* genes were distributed unequally on *P. trichocarpa* chromosomes, with 16 collinear gene blocks, indicating that whole/segmental-genome duplication plays an important role in expanding *PtSR* family genes.

Environmental stresses adversely affect plant growth and productivity, and plants have to evolve multiple biochemical and molecular mechanisms in response to various stresses for survival. Notably, plant SR proteins could function as central coordinators of responses to environmental changes. Stress signals can affect both the phosphorylation status and subcellular localization of some SR proteins [52]. Therefore, *SR* genes can participate in multiple abiotic stress and phytohormonal responses. In *Arabidopsis*, the *sr34b* mutant was sensitive to cadmium by regulating the *IRON*-*REGULATED TRANSPORTER 1* (*IRT1*) gene [53]. The *sr40* and *sr41* mutants displayed salt and ABA hypersensitivity [54]. The overexpression of *MeSR34* enhanced the tolerance to salt stress in transgenic *Arabidopsis* through maintaining ROS homeostasis and affecting the CBL-CIPK pathway [55]. We found that 70% of the *PtSR* genes were significantly differentially expressed under hormone and abiotic stress treatments in *P. trichocarpa*, while cold was the most obvious abiotic stress that affected the expression of *PtSR* genes. In addition, the results of qRT-PCR were not always consistent with the *cis*-elements analysis of the promoter region. For example, no stress responsive *cis*-element was found in the promoter region of *PtRS2Z32* (Appendix A), but qRT-PCR results suggested that *PtRS2Z32* was induced by cold, salt and MeJA treatments. This strongly indicates that there may be some unrevealed stress responsive *cis*-elements in regulating of the response to stress in *P. trichocarpa*. It also may be due to the regulation of *PtSR* gene on other stress-responsive genes.

Alternative splicing (AS) is an important mechanism in the regulation of gene expression, and this post-transcriptional process greatly enhances transcriptome and proteome complexity. Of note, *SR* genes can also be alternatively spliced, and participate in diverse life progresses of plants. For example, 45 transcripts were found from 18 *MeSR* genes under normal conditions, while 55 transcripts were identified under salt treatment [55]. In *Arabidopsis*, about 95 transcripts were produced from only 15 *SR* genes, thereby increasing the complexity of the *SR* gene-family transcriptome by six-fold [47]. In this study, we found that ~42% of the *PtSR* genes could be alternatively spliced in tissues and/or under abiotic/hormone stresses, and 45 transcripts were produced from the 24 *PtSR* genes. Under abiotic/hormone stress treatments, the AS patterns of some *PtSR* genes were altered, implying that the alteration of AS variants of *PtSR* genes may be related to the splicing-site selection or splicing-assembly changes in responses to stress. Moreover, no AS event of the RSZ subfamily genes was found in different tissues or under various stresses, which may be related to the fundamental roles of this subfamily of genes.

Different isoforms of a gene that undergo alternative splicing may play antagonistic roles in plant-growth and defense responses. For example, *PtrWND1B* could produce two transcripts (*PtrWND1B-s* and *PtrWND1B-l*) that played antagonistic roles in fiber cell-wall thickening [56]. In this study, about 42% *PtSR* genes changed their AS patterns under phytohormonal and abiotic stresses. There were significant differences in the number and abundance of *PtSR* transcripts expressed under different stresses or different treatment time points. Long transcripts of *PtSR* genes greatly accumulated under cold stress, suggesting that *PtSR* genes may play important roles in the response to cold.

It has been reported that *SR* genes participate in the alternative splicing of other genes and play important roles in stress responses. However, little is known about the function of *SR* genes in *P. trichocarpa*. We therefore investigated the role of *PtSCL30* in response to stress. We found that the overexpression of *PtSCL30* decreased the freezing tolerance of *Arabidopsis*, and the 35S::*PtSCL30* OE lines were hypersensitive to salt stress at the germination stage. These results indicate that *PtSCL30* may act as a negative regulator in cold and salt response by affecting the alternative splicing of the relevant genes. However, this is in contradiction with the positive role of the RS domains of *AtSRL1* and *AtRCY1* in the tolerance to salt stress [57]. This is likely because the overexpression of full-length (including the RRM and RS domains) or only the RS domains from *SR* genes would result in different stress phenotypes by inducing different gene-splicing outcomes. To further explore the genes that were regulated by the overexpression of *PtSCL30* in response to cold, we analyzed the RNA-seq results of the Col-0 and 35S::*PtSCL30* OE lines (OE10) under cold-stress (4 °C) treatment for 24 h. *PtSCL30* not only affected the expression of *Arabidopsis* genes, but also mainly acted as a splicing factor to regulate the alternative splicing of some *Arabidopsis* genes, such as cold-related genes (*CP29*, *ICE2*, *LHY*, and *COR15A*), which may result in the freezing-sensitive phenotype of the 35S::*PtSCL30* lines under cold treatment.

Our results indicate that *PtSR* genes may participate in multiple aspects of plant growth and development through their responses to various environmental conditions. Therefore, investigating the molecular mechanisms of different transcripts of *PtSR* genes in response to stresses may be important in analyzing the functions of *PtSR* genes, and it may also provide new insights into further functional elucidation of *SR* genes in woody plants.

## 4. Materials and Methods

### 4.1. Identification and Basic Features of PtSR Family Genes

The genomic sequences, coding sequences (CDS), protein sequences, and gene annotations of *P. trichocarpa* and *Arabidopsis*, respectively, were downloaded from Phytozome v13.1 [58] and TAIR10 [59]. To identify *PtSR* genes in *P. trichocarpa*, we aligned *Arabidopsis* SR proteins against *P. trichocarpa* proteins to retrieve RRM-containing homologs, and, based on the definition of the RS domain [25], we obtained a total of 24 PtSR-coding genes in *P. trichocarpa*. Their theoretical protein isoelectric points (pI), hydrophobicities, and the grand average of hydropathicities (GRAVY) were obtained using the ProtParam tool provided by ExPASy [60].

### 4.2. Phylogenetic Analysis, Domain Identification, and Architecture Visualization of PtSR Genes

The SR proteins of *Arabidopsis* and *P. trichocarpa* were aligned by MAFFT v7.427 [61] to construct a phylogenetic tree by the maximum likelihood (ML) method in IQ-TREE v1.6.10 [62]. The best-fit protein-substitution model was selected upon Bayesian information criteria (BIC) in Model Finder [63]. Based on the definition of the RS domain in plants [25], we developed a PERL script to obtain the location information of the RS domain. To investigate other possible conserved functional domains within SR proteins, we downloaded the hidden Markov models (HMMs) of the proteins’ domains from the Pfam database v31.0 and used hmmscan [64] to search PtSR proteins against the HMMs with an E-value cutoff of 1 × 10^−5^. Accordingly, we obtained the location information of coding exons and protein domains of the PtSR family, and visualized their gene exon–intron and protein-domain architectures with TBtools v1.0971 [65].

### 4.3. Chromosomal Localization and Expansion History of PtSR Genes

The chromosome locations of the *PtSR* genes were extracted from *P. trichocarpa* gene annotations. To reveal the expansion history of *PtSR* genes, we used MCScanX [66] to trace the duplications of *PtSR* genes. Then, the synteny analysis results were visualized using the Circos program implemented in TBtools v1.0971 [65]. In dating whole-genome duplication (WGD)/segmental duplication events, we used MCScanX [66] to search for collinear gene pairs in the genome of *P. trichocarpa* or between the genomes of *P. trichocarpa* and *Arabidopsis*. Then, the synonymous substitution rate (Ks) value of each pair of collinear genes was calculated using the YN method in KaKs_Calculator 2.0 [67].

### 4.4. GO-Term Enrichment and Promoter Cis-Element Analysis of PtSR Genes

We firstly assigned GO terms for *PtSR* genes using eggnog-mapper v2 [68] and performed the GO enrichment analysis using TBtools v1.0971 [65]. Bubble-plot representations of enriched GO terms were generated by an R script. Additionally, we extracted the 2-kb sequences upstream of the translation start site of *PtSR* genes and then predicted their regulatory *cis*-elements in PlantCARE [43].

### 4.5. Expression Profile Analysis of PtSR Genes

*P. trichocarpa* transcriptome datasets of root, stem, and leaf tissues were downloaded from the NCBI Sequence Read Archive (SRA) (accession ID: ERP021848) [69]. To obtain the expressions of *PtSR* genes in the tissues, we employed hisat2 [70] to map the reads on the corresponding genome and stringtie [71] to calculate the transcripts-per-million-reads (TPM values) of *P. trichocarpa*-expressed genes, including *PtSR* genes. We then extracted the TPM values of *PtSR* genes in the three tissues and regarded the other expressed genes as background ones. Accordingly, we used R programming to plot a heatmap of the expressions of *PtSR* genes and a boxplot comparing the difference in expression between the *PtSR* genes and the background genes, wherein the *P* value was calculated by the Wilcoxon test.

### 4.6. Quantitative RT-PCR (qRT-PCR) Analysis

Total RNA was extracted using RNAprep Pure Plant Kit (Polysaccharides & Polyphenolics-rich, TIANGEN, Beijing, China), and the extracted RNA was used as a template for first-strand cDNA synthesis by the PrimeScript™ RT reagent Kit (TaKaRa, Japan) with gDNA Eraser. qRT-PCR was performed using a CFX96 Real-time PCR Detection System with ChamQ SYBR Color qPCR Master Mix (Vazyme, Nanjing, China). The primer sequences for qRT-PCR were designed using Pimer3 (Appendix A). PCR reactions were performed under the following conditions: 95 °C for 3 min, which was followed by 40 cycles of 95 °C for 10 s, 58 °C for 30 s and 72 °C for 30 s. The relative expression level of each gene was analyzed using the 2^−∆∆CT^ method [72]. The constitutively expressed *Histone3* genes of *HIS-1* (*Potri.002G026800*) and *HIS-2* (*Potri.005G072300*) in *P. trichocarpa* were used as the optimal reference gene for normalization. Each sample was analyzed in triplicate.

### 4.7. Alternative Splicing (AS) Pattern Analysis of PtSR Genes

In order to analyze the AS patterns of *PtSR* genes in different tissues and various abiotic/hormone stresses, two-month old potted cultivations of *P. trichocarpa* seedlings, cultured in an artificial climate chamber at 25 °C and a photoperiod of 16/8 h light/dark cycle, were then treated under ABA (100 μM), SA (5 mM), MeJA (1 mM), cold (4 °C), drought (20% PEG6000), and salt (200 mM), respectively. For the hormone treatments, an aqueous solution of each hormone was sprayed on the leaves of the treated plants, and the fourth fully expanded leaves were collected after 24 h (h). For cold treatment, *P. trichocarpa* seedlings were cultured in an artificial climate chamber at 4 °C for a 16/8 h light/dark cycle over 7 days (d), and each fourth, fully expanded leaf was collected. For drought and salt treatments, aqueous solution of PEG6000 and NaCl were separately poured onto the soil, and the fourth, fully expanded leaf of each treated *P. trichocarpa* seedling was collected after the 24-h and 7-d treatments, respectively. Comprehensive analyses of the splicing patterns of *PtSR* genes were performed by RT-PCR. Gene-specific primers corresponding to the first and last exon (or second exon and penultimate exon) of each gene were adopted (Appendix A). The amplified products were resolved in 1.2–1.5% agarose gels, and the lines, in different sizes, were purified with the SanPrep Column DNA Gel Extraction Kit (Sangon Biotech). The alternatively spliced PCR products of *PtSR* genes were cloned into PMD^T^^M^-19 as vectors for sequencing.

### 4.8. Plasmid Construction and Plant Transformation

An 822-bp fragment of *PtSCL30* coding sequence was amplified with the *PtSCL30*-*Kpn*IF primer GGGGTACCATGAGGAGATATAGTCCACCACACT and *PtSCL30*-*Xba*IR primer GCTCTAGATCTAGCATGCCTTGGTGACAA. Then, the amplified fragment was cloned into a pCAMBIA1300-sGFP vector between the *Kpn*I and *Xba*I sites to construct the overexpression vector of *PtSCL30*. Next, the constructed *pCAMBIA1300*-*PtSCL30* plasmid was transferred into *Agrobacterium tumefaciens* GV3101-competent cells, and the transformed *Agrobacterium* was used for plant transformation. The floral-dip method was used for the heterologous transformation of *PtSCL30* into *Arabidopsis*.

### 4.9. Abiotic Tolerance and Seed Germination Assays

Three-week old wild-type plant Col-0 and three 35S::*PtSCL30*-overexpression lines (OE2, OE10, and OE16), grown in soil under a long-day photoperiod (16 h/8 h light/dark cycle) were used for the freezing-tolerance assay, as described by Jia et al. [73]. Briefly, the program was set at 4 °C for 10 min and 0 °C for 20 min and by decreased by 1 °C/h to the desired temperatures. After freezing treatment, the plants were grown at 4 °C in dark for 12 h, and then were warmed to 22 °C for an additional 3 d. The phenotype of each line was observed, and the survival rate of the lines was also counted. The sensitivity of seed germination to salt stress was assayed on 1/2 Murashige and Skoog (MS) agar plates supplemented with different concentrations of NaCl (0 mM, 150 mM, and 175 mM). Seeds were incubated at 4 °C for 48 h, and radicle emergence was used as an indication of seed germination. For salt and drought resistance assay on plates, Col-0 and 35S::*PtSCL30* overexpression lines were firstly germinated on 1/2 MS media under normal conditions, and one-week old seedlings with similar root length were transferred to 1/2 MS agar plates containing various concentrations of NaCl or mannitol.

### 4.10. High-Throughput mRNA Sequencing and Analysis

For RNA sequencing (RNA-seq), three-week old seedlings of wild-type *Arabidopsis* (Col-0) and an 35S::*PtSCL30* overexpression line (OE10), grown in soil at 22 °C, were treated at 4 °C for 24 h. Leaves of each line were collected, and total RNA was isolated by using the TRIzol reagent (Invitrogen, Carlsbad, CA, USA) following the manufacturer’s procedure. Then, the isolated RNA was quantified using a NanoDrop ND-1000 (NanoDrop, Wilmington, DE, USA), and the RNA integrity was assessed by Agilent 2100 (Agilent Technologies, Palo Alto, CA, USA). Finally, the isolated RNA was sent to Lianchuan Biotechnology Co., Ltd. (Hangzhou, China) to generate a cDNA library, and the constructed RNA-seq libraries were sequenced on an Illumina Hiseq 4000.

For RNA-seq analysis, the clean reads were aligned to the *Arabidopsis* genome and the reference transcripts (TAIR10) using hisat2-2.1.0 [70]. The gene expression was calculated using StringTie v2.0.3 [71]. Both the edgeR and DESeq2 programs [74,75] were employed to predict differentially expressed genes (DEGs), which satisfied the three criteria: the sum of the expression values (TMM) ≥ 5, log2 fold changes > 1 or < −1, and false discovery rate (FDR) < 0.05 between the compared samples. CASH [76] was finally used to detect the differentially AS genes (adjusted *p* value < 0.05) between the compared samples.

## 5. Conclusions

In this study, we identified 24 *PtSR* genes and their evolutionary history in *P. trichocarpa*, revealed their conservation and divergence of expression and AS in tissues and under abiotic/hormone stresses, and exemplified a cold-affected gene, *PtSCL30*, to explore its molecular mechanism and functions. The *PtSR* genes were divided into six subgroups, and they were distributed unequally on *P. trichocarpa* chromosomes with 16 collinearity gene blocks, which were generated by two expansion events of genome triplication and duplication before and after the divergence of *P. trichocarpa* from *Arabidopsis*. The protein domain architecture analysis showed that PtSR proteins were evolutionarily conserved splicing factors, and correspondingly, that *PtSR* genes were almost constitutively and abundantly expressed, and some were coupled with conserved AS in roots, stems, and leaves. Besides this, the expression levels of *PtSR* genes were significantly higher than the genomic background genes of *P. trichocarp**a*. The constitutive and abundant expression and AS of *PtSR* genes strongly suggested the conserved and fundamental roles of *PtSR* genes in different tissues of *P. trichocarp**a* in normal growth. The majority (~83%) of *PtSR* genes also responded to at least one of abiotic/hormone stresses (e.g., cold, drought, salt, SA, MeJA, or ABA), and, of these, cold stress led to a dramatic perturbation in the expression and/or AS profiles of 18 *PtSR* genes (~75%). In contrast to the static expression in normal growth, perturbation under abiotic/hormone stresses, especially cold stress, suggested the potential regulatory roles of *PtSR* genes in responses to stress. In support of this, the overexpression of cold-upregulated *PtSCL30* in *Arabidopsis* decreased the plants’ freezing tolerance, probably through AS changes of the critical cold-responsive genes (e.g., *ICE2* and *COR15A*) induced by *PtSCL30* overexpression. Together taking account that the transgenic plants were salt-hypersensitive at the germination stage, we have suggested *PtSCL30* functions as negative regulators in cold and salt stresses. These results would help decipher the screening and functional analyses of *PtSR* genes and may provide a foundation for further functional elucidation of *SR* genes in woody plants.

## Figures and Tables

**Figure 1 ijms-22-11369-f001:**
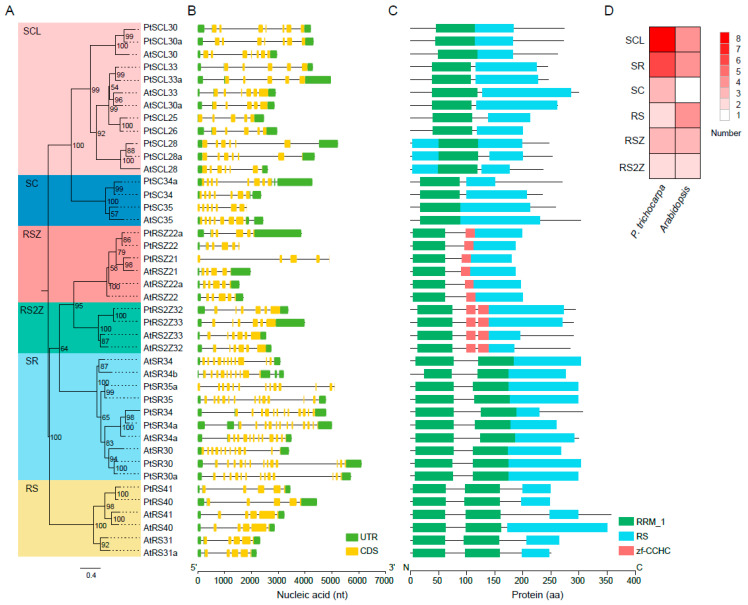
Phylogenetic relationships and exon/intron and domain architectures of *SR*-family genes in *P. trichocarpa* and *Arabidopsis*. Phylogenetics of *SR* genes. Multiple alignment of the *Arabidopsis* and *P. trichocarpa* SR proteins were performed by MAFFT to construct a maximum likelihood (ML) tree by IQ-TREE. (**A**) The ML tree was assessed by an ultrafast bootstrap with 5000 replicates, and bootstrap values greater than 50% are shown. The six clusters in shaded colors indicate the known conserved subfamilies (i.e., SCL, SC, RSZ, RS2Z, SR, and RS); (**B**) exon/intron structures of *PtSR* genes. UTR and CDS indicate the untranslated region and coding sequences, respectively; (**C**) protein domains of *PtSR* genes. The visualizations of exon/intron and protein-domain architectures were created by TBtools, using their gene- and protein-information datasets; (**D**) a heatmap showing the numbers of the six subfamilies of *Arabidopsis* and *P. trichocarpa SR* genes.

**Figure 2 ijms-22-11369-f002:**
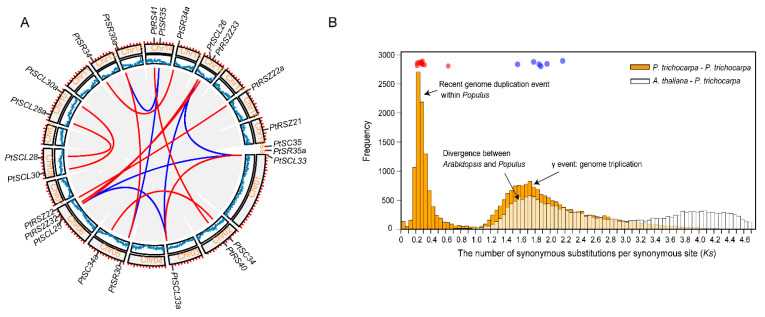
Chromosomal distribution and expansion events of *PtSR* gene family. (**A**) The chromosomal distribution and collinearity gene blocks the containing *PtSR* genes. The outer circle indicates *P. trichocarpa’s* 19 chromosomes (Chr) and scaffolds (s), marked with a distribution of *PtSR* genes; the middle circle indicates gene density on the corresponding chromosomes; and the inner grey curves indicate gene collinearity blocks between and within chromosomes, where the close paralogous pairs of *PtSR* genes are marked in blue or red curves, according to their expansion events; (**B**) the frequency of *Ks* values of the collinearity of gene pairs within the *P. trichocarpa* genome and between the *P. trichocarpa* and *Arabidopsis* genomes. The blue circles indicate the *PtSR* gene pairs generated by genome triplication event (i.e., γ) before the divergence of *P. trichocarpa* and *Arabidopsis*, and the red circles indicate the *PtSR* gene pairs generated by the recent genome duplication of *P. trichocarpa* after the divergence from *Arabidopsis*. The collinearity of the *PtSR* gene pairs and their *Ks* values are provided in Appendix A.

**Figure 3 ijms-22-11369-f003:**
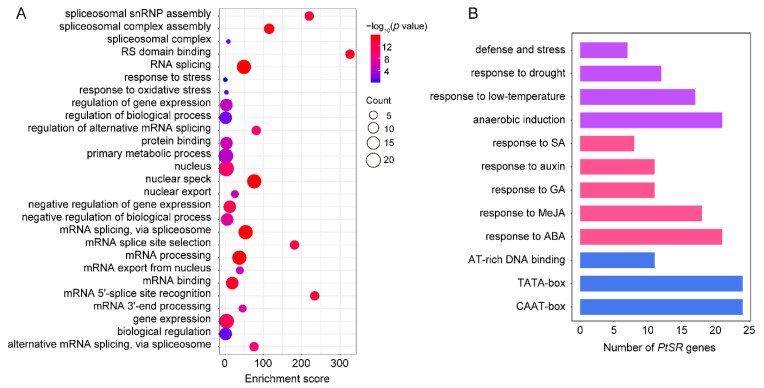
GO term enrichment and promoter *cis*-element analysis of *PtSR* genes. (**A**) GO term enrichments of *PtSR* genes. The dot sizes represent the numbers of enriched genes and the colored bars represent the significant levels of GO term enrichment; (**B**) the numbers of *PtSR* genes containing various *cis*-acting elements. Purple, red, and blue bars represent the *cis*-acting elements in response to abiotic stresses, phytohormones, and the fundamental core elements in *PtSR* gene promoters, respectively.

**Figure 4 ijms-22-11369-f004:**
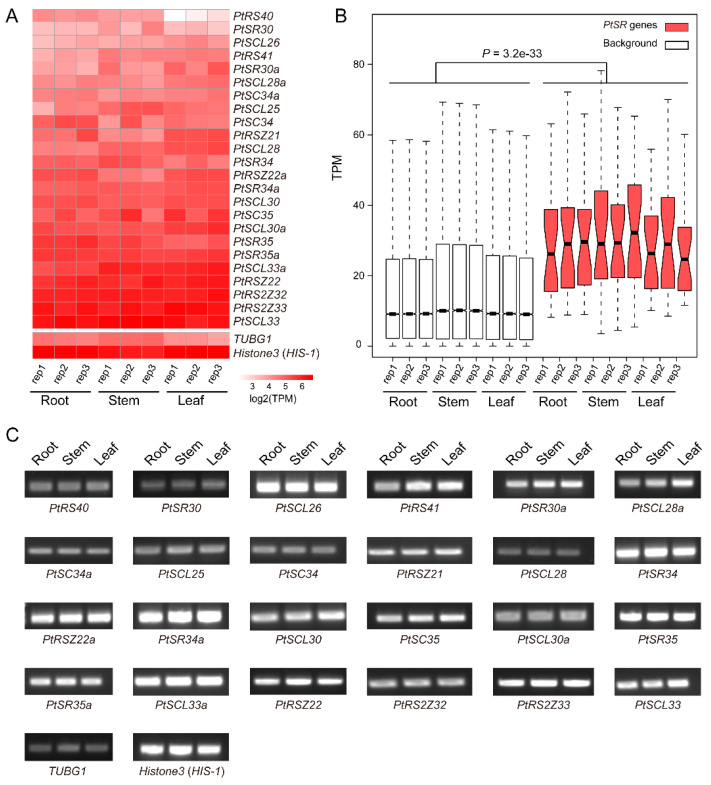
The expression profiles of *PtSR* genes in *P. trichocarpa* tissues. (**A**) A heatmap of the expression profiles of *Pt**SR* family genes in roots, stems, and leaves. Transcripts per million reads (TPM) was used to represent the expression of each gene and log2 transformed to generate the heatmap; (**B**) A boxplot showing expression differences between *PtSR* genes and background genes. Except for *PtSR* genes, the other genomic expressed genes were selected as background, and the differential level in expression between *PtSR* genes and the background was assessed by Wilcoxon test.

**Figure 5 ijms-22-11369-f005:**
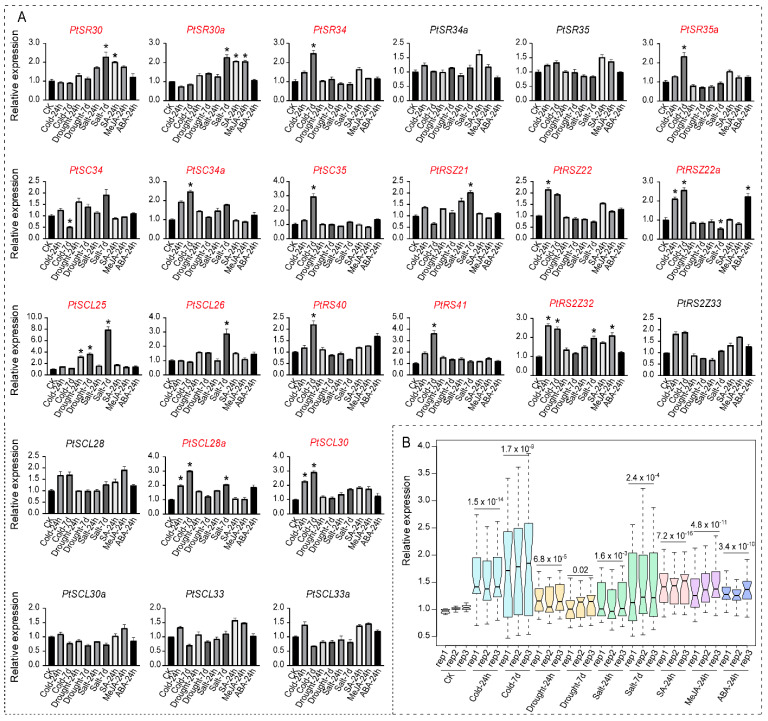
The expression profiles of *PtSR* genes under hormones and abiotic stresses. (**A**) The expression profile of all 24 *PtSR* genes. The red-colored *PtSR* genes represent the genes that were differentially regulated by at least one of the six stresses, and an asterisk above the bars represents a significant difference between treatments and control (* *p* value < 0.01 and factor of change >2). Error bars represent the standard deviations of three biological replicates; (**B**) A boxplot showing expression differences of *PtSR* genes between the treatments and CK control. The differential significance between the treatment and control of each *PtSR* expression was shown on the corresponding treatment. Relative expression level of *PtSR* genes were normalized with *Histone3* genes *HIS-1* (*Potri.002G026800*) and *HIS-2* (*Potri.005G072300*) under cold (4 °C), drought (20% PEG6000), salt (200 mM NaCl), SA (5 mM), MeJA (1 mM), and ABA (100 μM) treatments.

**Figure 6 ijms-22-11369-f006:**
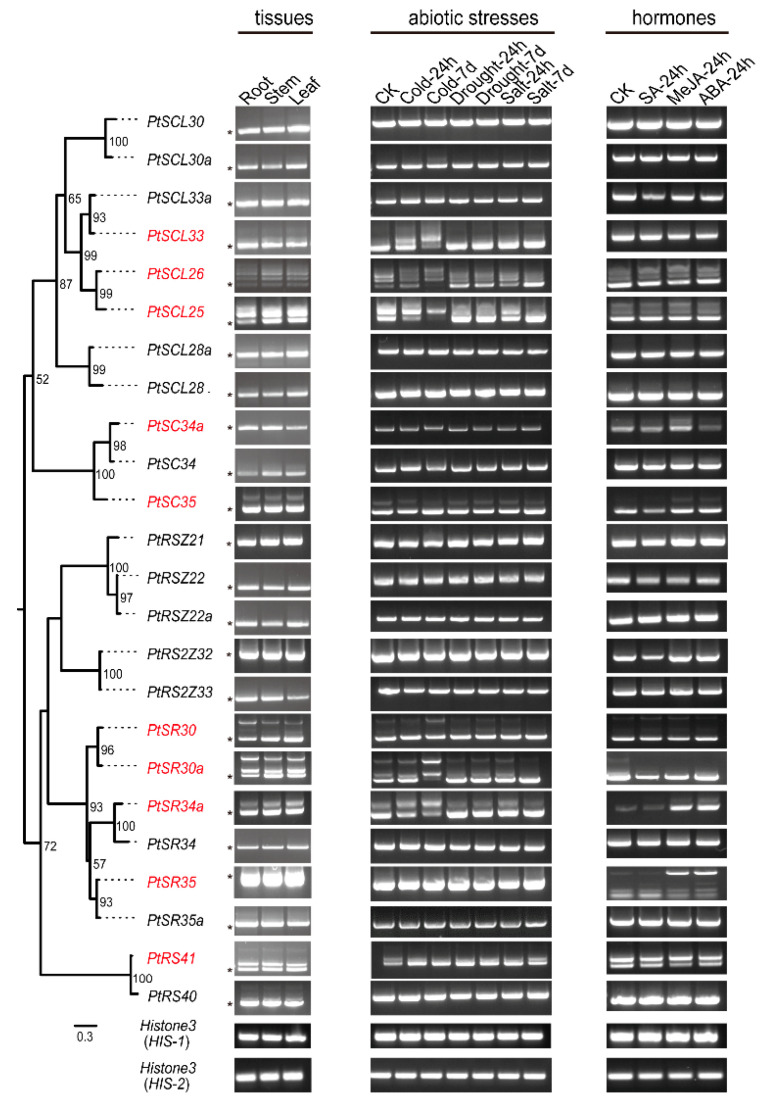
The alternative splicing of *PtSR* genes in *P. trichocarpa* tissues and under hormones and abiotic stresses. The left panel indicates the phylogenetics of *PtSR* genes. The AS patterns of *PtSR* genes were determined in the tissues (root, stem, and leaf) under normal condition and in leaf tissue under different stress treatments, including cold (4 °C), drought (20% PEG6000), salt (200 mM NaCl), SA (5 mM), MeJA (1 mM), and ABA (100 μM). The red genes labels represent genes that underwent AS events in the tissues and/or under abiotic/hormone stresses. Asterisks represent a basic transcript of the *PtSR* gene. Primers used for investigating alternative splicing isoforms of *PtSR* genes were provided in Appendix A.

**Figure 7 ijms-22-11369-f007:**
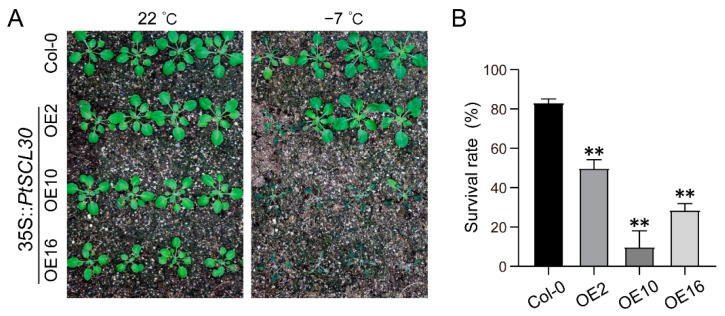
*PtSCL30* is involved in the freezing stress response. (**A**) Freezing tolerance of *PtSCL30* overexpression lines (OE2, OE10, and OE16) and wild-type plants (Col-0) after being exposed to −7 °C for 6 h; (**B**) the survival rate of Col-0 and 35S::*PtSCL30* OE lines after being exposed to −7 °C for 6 h. Asterisks indicate significant differences between Col-0 and 35S::*PtSCL30* OE lines (** *p* < 0.01).

**Figure 8 ijms-22-11369-f008:**
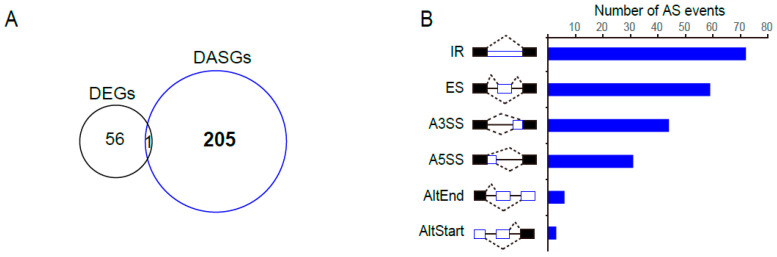
Overexpression of *PtSCL30* changes the alternative splicing profiles of genes under cold stress. (**A**) The number of differentially expressed genes (DEGs) and alternatively spliced genes (DASGs) in 35S::*PtSCL30* OE lines under cold stress; (**B**) the AS types and numbers induced by the overexpression of *PtSCL30*; (**C**) the GO-enriched network and the relevant DASGs. The big circles indicate the GO terms and the small circles represent the DASGs. The gene network involved in abiotic stress, including cold stress, is shown in red shadow; (**D**) four examples of DASGs after the overexpression of *PtSCL30* under cold stress from RNA-seq data. The RNA-seq coverage (black or blue) and junction reads (on the arcs) for the samples are shown above the reference genes. The differential AS loci are marked in red shadow with their AS type.

**Figure 9 ijms-22-11369-f009:**
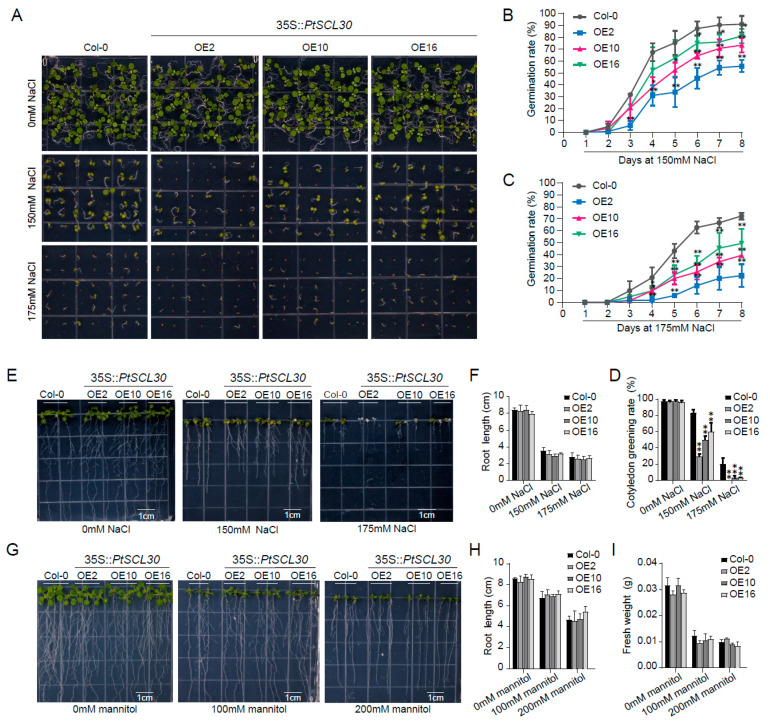
Salt and drought tolerance of *PtSCL30* transgenic plants. (**A**) Seed germination assay of Col-0 and 35S::*PtSCL30* OE lines growing in 1/2 MS medium supplemented with NaCl; (**B**,**C**) Germination rate of Col-0 and 35S::*PtSCL30* OE lines under salt treatments; (**D**) cotyledon-greening rate of Col-0 and 35S::*PtSCL30* OE lines under salt treatments; (**E**) salt tolerance assay of Col-0 and 35S::*PtSCL30* OE lines growing in 1/2 MS medium supplemented with NaCl; (**F**) root length of Col-0 and 35S::*PtSCL30* OE lines under salt treatments; (**G**) drought tolerance assay of Col-0 and 35S::*PtSCL30* OE lines growing in 1/2 MS medium supplemented with mannitol; (**H**) root length of Col-0 and 35S::*PtSCL30* OE lines under drought treatments; (**I**) fresh weight of Col-0 and 35S::*PtSCL30* OE lines under drought treatments. Error bars indicate the standard deviation. Asterisks indicate significant different levels between Col-0 and 35S::*PtSCL30* OE lines (* *p* < 0.05 and ** *p* < 0.01).

## Data Availability

The data that supports the findings of this study are available in the Appendix A at IJMS online. The transcriptome data of the Col-0 and 35S::*PtSCL30* overexpression lines (OE10) under cold stress have been deposited at the NCBI (PRJNA760989) and CNSA (CNP0002192).

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
