# Peer review of "The SR Splicing Factors: Providing Perspectives on Their Evolution, Expression, Alternative Splicing, and Function in Populus trichocarpa"

_ijms, 2021, doi:10.3390/ijms222111369_

Round 1

Reviewer 1 Report

The presented study "The SR Splicing Factors: Providing Perspectives on their Evolu-2 tion, Expression, Alternative Splicing, and Function in Populus 3 trichocarpa" clearly summarizes the current knowledge and provides rationale for the expreimental design which is sufficiently described within the manuscript. The results and the conclusion made are satisfactory provided. Therefore this reviewer recomment to accept this high quality manuscript for the presentation.

Author Response

The presented study "The SR Splicing Factors: Providing Perspectives on their Evolution, Expression, Alternative Splicing, and Function in Populus trichocarpa" clearly summarizes the current knowledge and provides rationale for the experimental design which is sufficiently described within the manuscript. The results and the conclusion made are satisfactory provided. Therefore, this reviewer recommends to accept this high quality manuscript for the presentation.

Response 1: Thanks very much for your positive comments.

Reviewer 2 Report

The paper is very interesting, research is well designed and well performed, but there are two weak points that must be adressed before being acceptable for publication.

1.-Conclusions: According to authors SRproteins are negative regulators of stress response, mainly salt and cold. This is in contradiction with previous reports, like:

Forment J, Naranjo MA, Roldán M, Serrano R, Vicente O. Expression of Arabidopsis SR-like splicing proteins confers salt tolerance to yeast and transgenic plants. Plant J. 2002 Jun;30(5):511-9. doi: 10.1046/j.1365-313x.2002.01311.x. PMID: 12047626.

An alternative explanation of the observed results, without involving a role as negative regulator, could be some kind of problems in the cell wall or in ion transport, that could affect the plant response to stress. Authors should comment this in the discussion and consider this possibility in the conclusion.

2.- There are not descriptions in mat and met of the Populus cultivation and sampling. How were plants cultivated? How was stress applied? How old were plants where samples were collected? Without this information reproducibility of the results is compromised. Please, give a detailed description of the experiments with populus plants.

Author Response

The paper is very interesting, research is well designed and well performed, but there are two weak points that must be addressed before being acceptable for publication.

1. Conclusions: According to authors SR proteins are negative regulators of stress response, mainly salt and cold. This is in contradiction with previous reports, like:

Forment J, Naranjo MA, Roldán M, Serrano R, Vicente O. Expression of Arabidopsis SR-like splicing proteins confers salt tolerance to yeast and transgenic plants. Plant J. 2002 Jun;30(5):511-9. doi: 10.1046/j.1365-313x.2002.01311.x. PMID: 12047626.

An alternative explanation of the observed results, without involving a role as negative regulator, could be some kind of problems in the cell wall or in ion transport, that could affect the plant response to stress. Authors should comment this in the discussion and consider this possibility in the conclusion.

Response 1: Thanks for this valuable suggestion. We cited this research and gave a potential reason for the different outcomes of stress response in the Discussion section (Lines: 458-462).

2. There are not descriptions in mat and met of the Populus cultivation and sampling. How were plants cultivated? How was stress applied? How old were plants where samples were collected? Without this information reproducibility of the results is compromised. Please, give a detailed description of the experiments with populus plants.

Response 2: In the Materials and Methods, we have added the relevant information you required (Lines: 536-546).